# A Novel Elderly Tracking System Using Machine Learning to Classify Signals from Mobile and Wearable Sensors

**DOI:** 10.3390/ijerph182312652

**Published:** 2021-11-30

**Authors:** Jirapond Muangprathub, Anirut Sriwichian, Apirat Wanichsombat, Siriwan Kajornkasirat, Pichetwut Nillaor, Veera Boonjing

**Affiliations:** 1Faculty of Science and Industrial Technology, Surat Thani Campus, Prince of Songkla University, Surat Thani 84000, Thailand; 6140320509@psu.ac.th (A.S.); apirat.w@psu.ac.th (A.W.); siriwan.wo@psu.ac.th (S.K.); 2Integrated High-Value of Oleochemical (IHVO) Research Center, Surat Thani Campus, Prince of Songkla University, Surat Thani 84000, Thailand; 3Faculty of Commerce and Management, Trang Campus, Prince of Songkla University, Trang 92000, Thailand; pichetwut.n@psu.ac.th; 4Department of Computer Engineering, School of Engineering, King Mongkut’s Institute of Technology Ladkrabang, Bangkok 10520, Thailand; kbveera@kmitl.ac.th

**Keywords:** machine learning, elderly tracking system, human activity recognition system, k-nearest neighbor, wearable sensors

## Abstract

A health or activity monitoring system is the most promising approach to assisting the elderly in their daily lives. The increase in the elderly population has increased the demand for health services so that the existing monitoring system is no longer able to meet the needs of sufficient care for the elderly. This paper proposes the development of an elderly tracking system using the integration of multiple technologies combined with machine learning to obtain a new elderly tracking system that covers aspects of activity tracking, geolocation, and personal information in an indoor and an outdoor environment. It also includes information and results from the collaboration of local agencies during the planning and development of the system. The results from testing devices and systems in a case study show that the k-nearest neighbor (k-NN) model with k = 5 was the most effective in classifying the nine activities of the elderly, with 96.40% accuracy. The developed system can monitor the elderly in real-time and can provide alerts. Furthermore, the system can display information of the elderly in a spatial format, and the elderly can use a messaging device to request help in an emergency. Our system supports elderly care with data collection, tracking and monitoring, and notification, as well as by providing supporting information to agencies relevant in elderly care.

## 1. Introduction

Elderly population growth is bringing significant challenges requiring more healthcare services and facilities [1,2,3,4,5]. Most elderly people have various health problems, such as memory problems, decreased ability to help themselves, fainting, or falling easily [6]. These problems cause the elderly to need care or monitoring by a caregiver [7]. In current social conditions, most caregivers leave the house for a career or to study. They are unable to constantly care for the elderly, and the elderly without care remains a problem. However, advanced technology can be applied to make a system for tracking and monitoring the elderly by using sensor technology, wireless networks, geographic technology, and smartphone devices [8]. Besides, these technologies have further capabilities and are nowadays ubiquitous. In the past, there have been systems for looking after the elderly in various forms [9,10,11,12,13]. Many studies have suggested that models of the human activity recognition system (HAR) tend to be best used to save the lives of the elderly [14,15,16,17]. The operation of the HAR system has five steps: (1) selecting and deploying sensors that are suitable for the human body or environment to capture user behavior or changes in the environment in which the user is doing activities, and combining two or more technologies or sensors will increase the efficiency of the tracking system [18]; (2) collecting and preprocessing data from sensors that are adapted for specific tasks; (3) separating useful properties from sensor data for later classification; (4) training a classifier by a suitable machine learning algorithm; and (5) testing the learning model to decide and report on the performance. Each of the steps above involves many technologies and available alternative methods, and there are also relevant research questions to manage [17,19,20]. These steps show that the development of the HAR system covers the use of detection technology, wireless networks, data processing, machine learning, degradation, classification, reduction, and other technologies following the model of the system. The HAR system can detect and monitor activities and hazards that may impact the elderly [5,8,21,22,23]. There are many devices used to receive activity data on users, which led to having large amounts of data. Thus, many machine learning methods have been employed in the HAR system to reduce and extract summaries from the data to reflect the entire data without circulation [24,25]. Thus, this work applied machine learning to discover patterns in activity data and then made predictions on the daily life activities of elderly to detect and analyze trends and help solve problems for elderly persons.

Many approaches of machine learning are widely and successfully used in human activity monitoring and elderly health care systems  [26,27]. For an example, SVM (support vector machine) and random forest [7] have been used to build models for activity prediction. Deep learning approach  [3,24,25] is popular in processing signal features based on personalized characteristics from elderly patients. It can extract automatically features from raw data, which can substitute for manually designed feature extraction procedures. However, deep learning is still confronted with reluctant acceptance by researchers owing to its abrupt success, bustling innovation, and lack of theoretical support. Therefore, it is necessary to demonstrate the reasons behind the feasibility and success of deep learning in human activity recognition despite the challenges [26]. Surveys of using machine learning in HAR system are found in prior literature [26,27]. Although machine learning has been employed in HAR systems, this field still faces many technical challenges because of the different input characteristics, such as activities of interest or different physical characteristics of humans that cause challenges. Moreover, the care and assistance of the elderly requires further information to support efficiency. Thus, these works compared popular machine learning algorithms to classify activities in order to find the most efficient algorithm for classifying human activities.

This paper focuses on developing a comprehensive system for tracking elderly information in many areas because caring for the elderly who are healthy or strong is still necessary to track personal information or welfare that should be received. The elderly who have an underlying disease or are at risk of daily life accidents should have a system that supports requesting emergency help. Furthermore, falling or fainting may cause the elderly to lose consciousness or even to die [1,28], which our system will automatically detect and alert the caregivers for immediate help. Besides, each elderly may have different movement characteristics with each activity. The activity classification model should be able to be amended or improved for each user, which affects choosing a machine learning platform that will be used to develop the system appropriately. In this paper, we show the entire process of developing the elderly tracking system that covers tracking personal information, activities, geolocation, fall notifications, and requesting emergency assistance. This system uses a mobile phone and the development of wearable sensor devices for tracking in both indoor and outdoor environments. The wearable sensor device is developed with sensor technology, and consideration of cost is an essential factor. Reduced costs mean fewer sensors, so more complex inference processes should be used to provide complete sensor data and accurate analysis. For this reason, we show the process for choosing the machine learning model suitable for activity classification and applying it to real-time activity classification. Furthermore, this study used a cooperative research model so that the developed system responds to user needs as much as possible as regards the elderly and related agencies. They have been involved in system planning and testing.

The rest of the paper is organized as follows. Section 2 explains the critical theories used in this research. Then, we present the research methodology in Section 3. This section has two sub-sections: system design and implementation. Section 4 shows the experiment and evaluation. Finally, the conclusion is provided in Section 5.

## 2. Background

### 2.1. Human Activity Recognition System

Human activity recognition (HAR) systems have been developed to protect human safety [29,30,31]. As an example, a system can monitor the movements of patients in a hospital [32,33,34]. In industries or factories, it is often used to monitor the health of workers in hazardous areas [35,36]. For healthcare, the system is designed to remember and record movement or behavior, as can be seen from exercise products [37,38]. The format of the HAR system depends on the data measured and on where the developer wants to install the sensors. The HAR systems can be divided into three forms according to sensor development: the ambient sensor-based HAR (ASHAR), the wearable sensor-based HAR (WSHAR), and the hybrid sensor-based HAR (WSHAR). ASHAR is a system that predicts human activities from sensors attached to the surrounding environment, on walls, doors, floors, and appliances. This system uses sensors to measure light, smell, frequency, and temperature. The operation will not bother users because the system does not have to be attached to the body; the price is low, but the system can be difficult to install or use [39,40,41,42,43]. WSHAR is a system format that identifies activities using sensors that are designed to be wearable, in which the activity classification uses machine learning [44,45,46,47]. This type of system can be used in a wide area, for example, employing watches, clothing, or devices designed specifically for a specific location [48,49,50,51,52]. The system may use small and sophisticated sensors, as well as high-cost equipment, and may disturb the daily lives of users. ASHAR and WSHAR have different strengths and weaknesses. Therefore, their combination makes them more accurate and more suitable for use. HSHAR is a system that combines two types of sensors in one system, with environmental detection and wearable sensors. This system is the most appealing form because of the low cost and flexibility of daily use [19,53].

The increase in the elderly and those who live alone leads to the need to develop a monitoring system and the need for new ways to solve problems [54]. The latest advancements in sensor technology make HAR usable for the elderly monitoring application [55]. HAR is used in the health care system and is extensively used to monitor the activities of the elderly in rehabilitation centers for chronic disease management and disease prevention [56]. HAR is also integrated into the smart home to monitor the daily activities of the elderly [57,58]. Furthermore, HAR is used to track patients at home, including in calorie consumption estimates to help prevent obesity [59].

### 2.2. k-Nearest Neighbor Algorithm

The k-nearest neighbors (k-NN) is one of the best known and most effective methods for supervised classification and should be one of the first options for classification studies when there is little or no previous knowledge about data distribution [60,61]. This algorithm classifies new cases by examining the k closest cases with known class labels. The class label with the most votes from those k closest neighbors is assigned to the new data point. The choice of the closest labeled cases depends on what distance measure is applied to features in the data. This method is suitable for numeric data, but variables with discrete values can need proper management. After that, we can combine all attribute values that can be measured to distances between cases. After calculating the distances between conditions or cases, we have chosen a set of conditions that classify as the basis for classifying new conditions. Finally, we determine the (near optimal) count *k* of the neighboring points. The number *k* should be odd in binary classification to ensure that one class label wins the vote. This algorithm is easy to use and can be applied to data analysis in many areas, for example, in a fall detection system [62,63] and in human activity recognition (HAR) [64,65].

### 2.3. Distance Metrics

Distance metric is the main criterion in clustering, and one could choose either Euclidean or Manhattan distance. Manhattan and Euclidean are popular for quantifying similarities between data items of a cluster. This work applied the Euclidean and Manhattan metrics to assess performances in activity classification by k-NN approach, to select the best alternative.

The euclidean distance or euclidean matrix method [61,66,67] is the common distance between two points along a straight line and is derived from the Pythagorean Theorem. The calculated values are referred to as the Euclidean norm. Early writings referred to this measurement as the Pythagorean distance.

Euclidean distance between two points **p** and **q** is the length of the line segment $\bar{\mathrm{pq}}$ if **p** = ($p_{1},p_{2},...,p_{n}$) and **q** = $(q_{1},q_{2},...,q_{n}$). In Cartesian coordinates, these are two points in the Euclidean n-space. Then, the distance (d) from **p** to **q**, or from **q** to **p** is determined by the Pythagorean formula:(1)$$\begin{array}{cc} d(p,q) & =d\left(q,p\right)=\sqrt{{\left(q_{1}-p_{1}\right)}^{2}+{\left(q_{2}-p_{2}\right)}^{2}+\cdots +{\left(q_{n}-p_{n}\right)}^{2}}\\ \; & =\sqrt{(\sum_{i=1}^{n}{\left(q_{i}-p_{i}\right)}^{2}}, \end{array}$$

The position of a point in the Euclidean n-space is a Euclidean vector. Therefore, **p** and **q** may be expressed as a Euclidean vector starting from the origin of the space (initial point) with the ends (terminal points) at two points, and the Euclidean norm or the Euclidean length or the size of the vector is
(2)$$\left\|p\right\|=\sqrt{p_{1}^{2}+p_{2}^{2}+\cdots +p_{n}^{2}}=\sqrt{p\cdot p},$$

The description of the vector is the element of the line from the origin of the Euclidean space (vector tail) to one point in that space (vector tip). Its length is the distance from the tail to the tip. The Euclidean norm of a vector is the Euclidean distance between the tail and the tip. The relationship between point **p** and **q** may be related to direction. For example, from **p** to **q** the vector is defined by
(3)$$q-p=(q_{1}-p_{1},q_{2}-p_{2},\cdots ,q_{n}-p_{n}),$$

In two- or three-dimensional space (n = 2, 3), this can be represented by an arrow from **p** to **q**. In any space, it can be considered the position of **q** relative to **p**. It may be called a displacement vector. If **p** and **q** are two positions of some of the moving points, so the last equation relates to the product of the points as the magnitude of the vector **p** from the origin. The Euclidean distance may also be given as follows, using the dot product or inner product:(4)$$\left\|p\right\|=\sqrt{\left(p-q)\cdot (p-q\right)}=\sqrt{{\left\|p\right\|}^{2}+{\left\|q\right\|}^{2}-2p\cdot q}.$$

Another common distance for Euclidean spaces and low-dimensional vector spaces is the Manhattan distance, which measures distances only along the axial directions. It is defined as
(5)$$\begin{array}{cc} d(p,q) & =d\left(q,p\right)=|q_{1}-p_{1}\left|+\right|q_{2}-p_{2}\left|+\cdots +\right|q_{n}-p_{n}|\\ \; & =\sum_{i=1}^{n}\left|q_{i}-p_{i}\right|. \end{array}$$

## 3. Methodology

### 3.1. System Design

The proposed elderly tracking system using machine learning was designed as shown in Figure 1.

In Figure 1, the proposed system consists of three components: tracking system, data management and machine learning, and application for related persons. The first component involves technologies that are applied to track the elderly both indoors and outdoors. The second component has the management of data derived from the tracking system. The final component applies the collected and analyzed data, and informs caregivers or related persons.

The tracking system in the first component and includes two parts. The outdoor part applies a global positioning system (GPS) sensor embedded in a smartphone and collects personal data of the elderly with web application-based questionnaires. For an indoor environment, we develop a wearable device with sensor technology. The proposed wearable devices use accelerometer and geolocation sensors to collect movement data and track the geographical location of the elderly. The tracking information from the first component will be managed by using the process in both devices and send to the next component.

The second component is data management and machine learning on the server, controlled with the web application. This component is divided into three main parts according to the workgroup: server, machine learning, and backend.

First, the server is responsible for storing program files, data, and processing. Users can manage data through web applications, but some of the processes are automatic in the form of data-driven operations. The data received are processed, passed on to the classification, and recording functions in the database. Initially, part of the data will be used to build the learning model using machine learning, training with 11,000 records that are described in more detail in Section 4.1. This obtained dataset is split into two groups: training and test sets of 80% and 20%, respectively. Both groups will continue to be used in machine learning part.

Second, machine learning in this component performs classification of the activities of the elderly. This part receives the sensor data and then adjusts the format to suit the activity classification model. We compare the models of machine learning used to classify activities using data from the wearable devices we developed in the first part to find the most efficient algorithm, as in Figure 2. From this experimental comparison among machine learning models shown in the figure, the k-NN was selected to activity classification in the next step due to its best performance. Next, data were classified into activities of the elderly using the k-NN classifier model, and the activity label was returned to the server. Finally, the backend will describe the selection, development, and use of models.

In this work, we tested the classification of 9 activities (more details in Section 4.1), such as walking, running, sitting, standing, and sleeping with ten models using the R program, as shown in Figure 2. Besides, we collected data from the right hip and right hip position to find a suitable position for use. The graph in Figure 2 shows that the k-NN (k-nearest neighbors) classifier was the most accurate in classifying activities and that the right hip position is suitable for use. After choosing a model, we developed the model to run on the server in the form of a web application. The selected model is applied to develop a web application to the user. The classification settings of the system are designed to be flexible for increasing activity types, changing the k value, or training information updates. The flexibility of this classification makes it always possible to improve classification efficiency and to apply the system to a wide variety of activities. The system deployed will predict the new activities of elderly from the models learned using the provided dataset.

The last component is an application for persons related to the elderly, such as grandchildren, elderly caregivers, or elderly care agencies. Those related persons use the web application to track and monitor the activities of the elderly. Furthermore, the system sends emergency alert SMS (Short Message Service) to elderly caregivers and agencies that the elderly need. Emergency alerts will be sent in the event of a fall or if the elderly press the emergency button on the wearable device. Access to the elderly information on the web application differs by type of user, according to their duties. For example, the grandchildren can only access the elderly in care, while the elderly care agencies can access information on all the elderly in the community.

### 3.2. Implementation

The system developed had two parts: hardware (wearable device) and software (web applications, commands on the server, and commands on wearable devices). The first part is the development of wearable devices for tracking activities and geographic locations of the elderly with sensor technology. We designed devices with critical components such as microcontrollers (NodeMCU), sensors, GPS modules, batteries, buttons, speakers, and charging modules in the box of 9 × 6.5 × 2.5 cm size. The tracking device is shown in the photo of Figure 3.

The GPS module and GY-521 module are applied to receive the personal activities data. The GPS module uses signals from satellites and then calculates the geographical position (latitude, longitude), sea-level elevation, and movement speed. The GY-521 module combines the capabilities of many sensors, including angular acceleration measurement, angular velocity measurement, angle measurement, and temperature. The system uses the data from the GY-521 module for activity classification and the data from the GPS module to identify the geographic location of the elderly. Furthermore, the device has a button for sending emergency help messages to the telephone number specified by the elderly in settings. In this case, we use the battery capacity of 2000 mAh and 3.7 volts. The tracking device continuously sends data once every second. Users can use it for approximately 15 h on a single charge. The NodeMCU with ESP8266 is a microcontroller that comes with esp8266, which can connect to a Wi-Fi network. It is necessary to set up the connection of the device to home Wi-Fi or smartphones of the elderly. The components of the tracking device are packed in the box, as shown in Figure 4.

The elderly wear the device on the right hip, and it must be turned on for it to work. The device will search for the internet network that is set up and then will continue to work sending and receiving data automatically. However, a limitation regarding internet connection of the developed wearable device is its design to automatically connect with home Wi-Fi or smartphone as predefined in setup, but it cannot connect to any other Wi-Fi.

As for software development, we use the Arduino IDE to develop a set of instructions for tracking devices. This set of instructions controls receiving data from sensors, process data, and send data to the server. The process is summarized in the flowchart in Figure 5.

The workflow diagram in Figure 5 was developed into a C-language instruction set and uploaded into the device. This diagram shows the workflow that begins with the tracking device trying to connect to the internet. Afterward, it will enter the loop that will check the data by pressing the emergency request button and the data from the sensor. If the information received is correct, the system will connect and send these data to the database. These processes use LED lights to communicate with users, illuminating or extinguishing each lamp according to the definition above of the tracking device.

Another essential part of software development is web applications and commands on the server. This part controls the working and decisions of the system. We develop web applications using PHP, HTML, JAVASCRIPT, and using SQL for database management. Furthermore, external systems are used to increase the efficiency of data display and more comfortable operation. Google Maps API is used to develop online maps that display spatial data with adjustable views, SMS services for sending notification messages, Google Chart API for creating graphs that can be updated in real-time. The system has different functions and access to information for each group of users since the elderly information is personal information that cannot be shared. Administrators or local authorities have access to all the elderly information in their area of responsibility. However, an elderly caregiver or elderly can access the elderly information of them only, with the interface structure shown in Figure 6.

## 4. Experiment and Evaluation

### 4.1. A Case Study

The study area was Moo 5, 6, 7, and 8, in Makham Tia Subdistrict, Mueang Surat Thani District, Surat Thani Province, Thailand. This area was chosen because of the validation of experimental data. Moreover, the proposed system arises from the needs of stakeholders involved in elderly care. There is the village headman, village volunteers, sub-district service organization, sub-district hospital, and Ministry of Social Development and Human Security. The total area of the four villages in 2115 hectares, the population is 1640 persons, with 807 males, 833 females, and 328 elderly persons (60 years or more). This study randomly sampled the elderly data of 240 persons, including 112 males and 128 females. The general information on gender, age range, blood type, education, work, and chronic diseases is summarized in Figure 7.

The data from above study area due to tracking system in outdoor part. The system uses online maps to present data for the elderly, allowing the data to be viewed in a spatial format. Moreover, there are tools in Street View and navigation that help to find the location of the elderly and make it easier to get there. Furthermore, the activity data of the elderly using tracking devices are summarized and displayed using graphs and symbols for an easier understanding of the user. Examples of the screen data display are shown in Figure 8.

The system has been developed to support all devices that can use the web application so that users can use their smartphone to access the system. A smartphone also makes it easy to take pictures and receive geolocation to save the elderly information. Furthermore, the emergency message system will work at full efficiency if the receiver opens the message with an internet-connected smartphone.

From the study area in outdoor part, we found that the related persons (elderly or caregiver) can monitor and track the elderly through the elderly’s mobile. They can get the position of an elderly who goes outside their home. However, there is limited availability of outdoor tracking system using GPS sensor embedded in smartphone and it has a low positioning accuracy. Moreover, it cannot identify the elderly activities performed. Thus, this work fulfilled these requirements by designing wearable devices for monitoring and tracking the elderly indoors. The presented wearable device was used to collect and send data to the server side. Initially, we used the wearable devices to collect data to create the dataset for building the learning model to predict elderly activities. Thus, the data were collected from 2 experienced persons and 10 elderly persons for 9 activities over 7 days every 5 min while wearing the device. Next, the data were properly prepared and preprocessed with data cleaning. The results from data preprocessing are shown in Table 1.

### 4.2. The Elderly Tracking System Using Machine Learning

Machine learning was used for the activity classification of elderly people in daily life. Alternative classification algorithms were trialed, and the k-NN model was selected, as discussed earlier.

In this study, the distance of k-NN method is calculated with two ways: Euclidean and Manhattan to compare and select the best performance of them. We use dataset from collected 9 elderly activities with 11,000 records, with more details given in Section 4.1. The dataset was split into two groups: training and test sets of 80% and 20%, respectively. The training set was used to build the learning model while the test set was used to validate the trained model. The performance results as shown in Table 2.

From Table 2, we found that the Euclidean distance is better than Manhattan distance for performance assessment. Thus, the Euclidean distance was selected for assessing the k-NN method. Next, the k-NN model was trained to label activity types of the elderly in daily life. The proposed k-NN algorithm is summarized in Algorithm 1.

**Algorithm 1:**The k-NN algorithm for activity classification

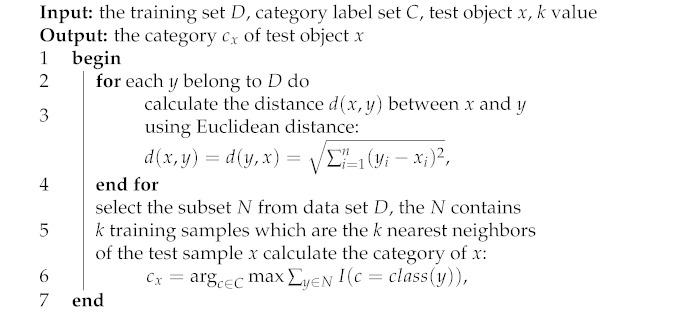



The data analysis by using KNN algorithm had training data stored in the server, which has large processing capacity and memory. The server side can analyze sensor data sent from wearables and quickly respond to data requests in real-time to clients. This algorithm, in Algorithm 1, based on brute force method will calculate distance from new point to every point in training data, sort distances and take k nearest exemplars, and then do a majority vote. Thus, the time complexity is estimated with O(k×n×d), where *n* is the number of points in the training dataset, *d* is data dimensionality, and *k* is the number of neighbors that we consider for voting.

In Algorithm 1, we use this algorithm to classify the data received from our wearable devices. The input of the algorithm consists of training data sets (*D*) covering all types of activities required, category of activity classes (*C*) paired with training data sets, *k* values, and data required to classify activity classes (test object *x*). The output is the class label or type of activity derived from the classification ($c_{x}$). The k-NN algorithm finds the distances between test data (test object *x*) and each data point (*y*) in the training set (*D*), using the Euclidean distance. Data from wearable devices for activity classification consists of 9 factors, which are 3-axis angular acceleration (A-X, A-Y, A-Z), 3-axis angular velocity (G-X, G-Y, G-Z), and 3 angles (Angle-X, Angle-Y, Angle-Z). An example of data for walking activity is shown in Figure 9. Therefore, the value n in the Euclidean distance equation of this algorithm is nine or the Euclidean distance equation for 9-dimensional space. The data with the closest distance will be selected for *N* numbers, in which the selected amount depends on the assigned *k*. The final step is to find the argument maximum of the class value from the selected data. The important thing about using this algorithm is that it should have suitable training data size and *k* value.

In the case study, we tested the elderly tracking system with ten volunteers, aged 62 to 73 years old, three males and seven females. Moreover, we use 2 experienced persons for activity movement that describe the dataset in Section 4.1. The test collected movement data for a total of 9 activities: sitting, standing, walking, running, falling, laying on one’s back, laying face down, laying on the left side, and laying on the right side. We used 11,000 data records as training data to find the appropriate *k* value and to test the accuracy of the model. The critical thing to note with the k-NN algorithm is that the number of features and the number of classes are not involved in determining the value of *k*. The selection of *k* is essential. When the *k* value is small, it means that noise will have a larger effect on the result. Large value makes calculations slower and opposes the basic philosophy behind the k-NN algorithm (those points that are near might have the same class). We used k-fold cross-validation [68,69,70], also known as k-fold cv, to determine the best *k* value. This is commonly done to set user-chosen parameters in a machine learning model during training. In this test, we divided the data into 10 test sets, *k* values from 1 to 50 were tried, and misclassification errors were the criterion to minimize. The test results show that the optimal *k* value is 5, seen in Figure 10.

The average accuracy of the k-NN model with *k* = 5 was 96.40%. We apply the algorithm with the system to classify the activities of the elderly in real-time. The system will work automatically to collect data, classify activities, and respond to classified activities. The algorithm of the work process in the overview is shown in Algorithm 2.

Algorithm 2 shows the overall system operation and decision-making. These processes begin with information sent from a wearable device (Figure 5), which is SOS.data and device.data. The SOS.data are used to check for emergency help requests by pressing a button on the wearable device and sending information for help. Some device.data (category $c_{x}$) are used for classifying activities by sending data to the classifier (Figure 7). If the activity class received is a crash activity, the system will send a notification to the assigned telephone number. The location.data are a user’s geolocation data, allowing tracking for emergency requests. The system uses location.data sent from wearable devices, but if the device is in a condition that it cannot receive location data, it will be extracted from personal data of the elderly instead (home location). The entire data analysis process ends at saving the data to the system database. Namely, if wearable device has some problem, the location from GPS from smartphone in outdoor part will be checked to get current location. However, the worst case of both indoor and outdoor is to lose the signal at the same time, then first the default assumption is home location before finding the current location.

**Algorithm 2:** The elderly monitoring algorithm.

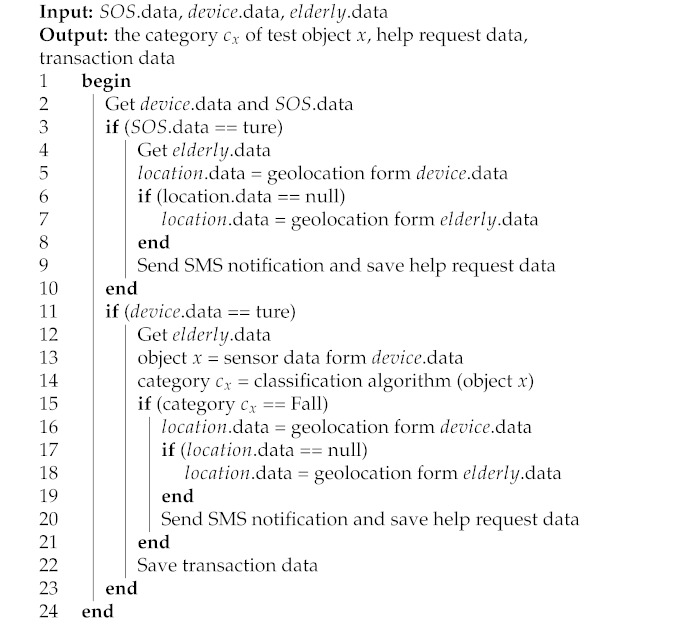



The data from the tracking device are matched with the data of the elderly who use the device. The system classifies activities and records in the database, which can be monitored in real-time via the web application. The monitoring screen shows the status of device information, users, device activation (online/offline), and recent activities. Users can view a summary of past activities of the elderly. The system shows the duration of each recorded activity, which helps to understand the activity in the daily life of the elderly. Furthermore, the geographical location data from the device is used to identify the elderly position on an online map. The system uses this information in conjunction with the geolocation of devices that enable web applications to enable navigation from users to the elderly. Examples of these screens are shown in Figure 11.

Regarding emergency cases, we designed the system to respond to activities that are expected to be harmful to the elderly. When the classification system finds a fall activity, it will send a notification message to the phone number of the caregiver in the form of SMS. The caregivers who receive the information can immediately check the information in the system or call back to check the safety of the elderly. Furthermore, the system has a function to request emergency assistance by pressing a button on the wearable device. This function sends a help message to the caregivers or agencies involved in caring for the elderly in the community. The information sent will help identify relevant information for the elderly and encourage more comfortable and faster access to help. The information includes personal information, disease and personal medicine, photos, and directions to the location of the elderly, as shown in Figure 12.

For system evaluation, we used questionnaires to assess user satisfaction and get recommendations for improving the system. Moreover, we interviewed the users testing the proposed system. We found that over 50% of users were satisfied at the highest level with the overall system usage, followed by 40% of users satisfied at a high level. Regarding the tracking devices, 70% of users were satisfied at a high level with the design that is easy to use and not complicated. For the designed wearable device, the weight is about 100 g. The data transfer from this device to the server is 150 bytes per second. However, a recommendation from users is to reduce the size of these devices. These recommendations are challenges for future system improvements, which can be done when essential components are developed to a smaller size with microcontrollers, batteries, or sensors.

## 5. Conclusions

This paper presents an activity tracking system for the elderly using machine learning, which can perform tracking in both indoor and outdoor environments. We developed web applications for data collection and managing data through mobile devices, and developed an activity tracking device in wearable form with sensors and microcontrollers. This device detects movement of the elderly for use in classification of activity with machine learning and identifies geographic information for monitoring the geolocation. In this case study, we tested the system and tracking device with data from elderly volunteers in four villages of Surat Thani province, Thailand. The results indicate that the k-NN classifier with k=5, and the attachment of the device to the right hip were the most suitable choices. The developed algorithm uses nine factors from wearable devices to classify nine activities with an average accuracy of 96.40% (sit, stand, walk, run, fall, lay on one’s back, lay face down, lay on left side, and lay on right side). Moreover, we have designed the system in terms of model settings to be flexible in adjusting the classification settings. The collaboration of devices and web applications makes it possible to use real-time elderly activity classification, in which the system can monitor the activities of the elderly and provide a summary report of daily activities or a notification in emergency case. Elderly caregivers can track the location of the elderly on an online map with a route guidance system to the elderly. The system also displays information summaries in graphs and data on online maps for an added perspective to help understand the data. From the system satisfaction questionnaire, more than 50% of the users had the highest level of satisfaction and 40% had a high level of satisfaction. However, the feedback from most users requires smaller sized wearable device, which is a challenge for future developments. This research has received a lot of attention and cooperation from community agencies because they can use the system for elderly tracking and use the information to support public policy making regarding the quality of life development of the elderly.

## Figures and Tables

**Figure 1 ijerph-18-12652-f001:**
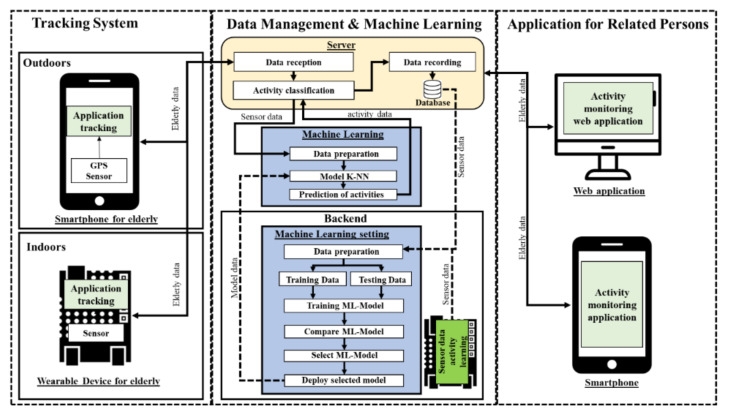
The proposed system architecture.

**Figure 2 ijerph-18-12652-f002:**
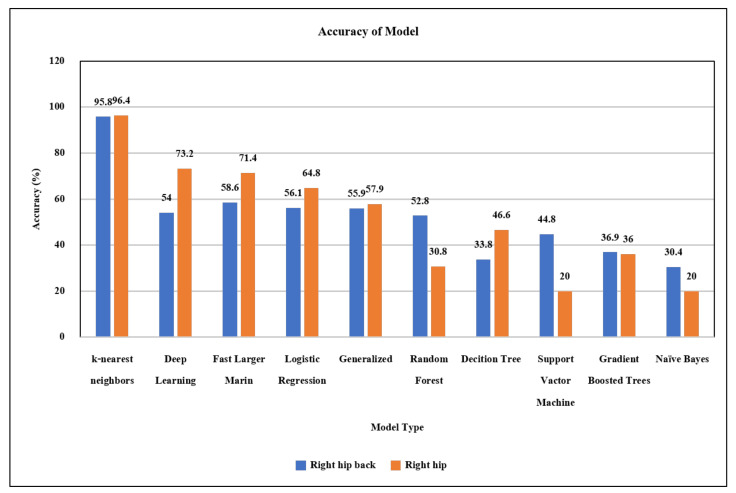
Comparison of classifiers with two alternative positions of the wearable device.

**Figure 3 ijerph-18-12652-f003:**
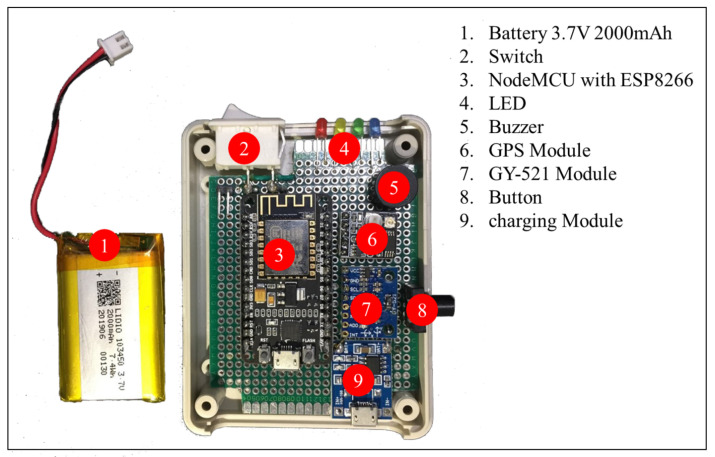
The internal characteristics of the tracking device.

**Figure 4 ijerph-18-12652-f004:**
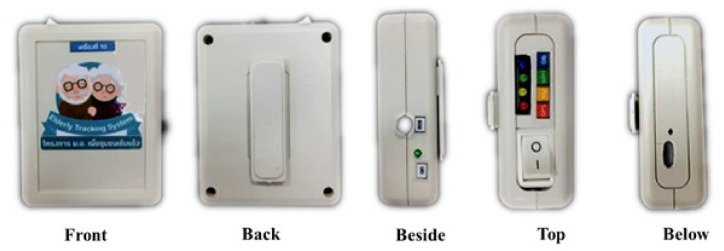
The external appearance of the tracking device.

**Figure 5 ijerph-18-12652-f005:**
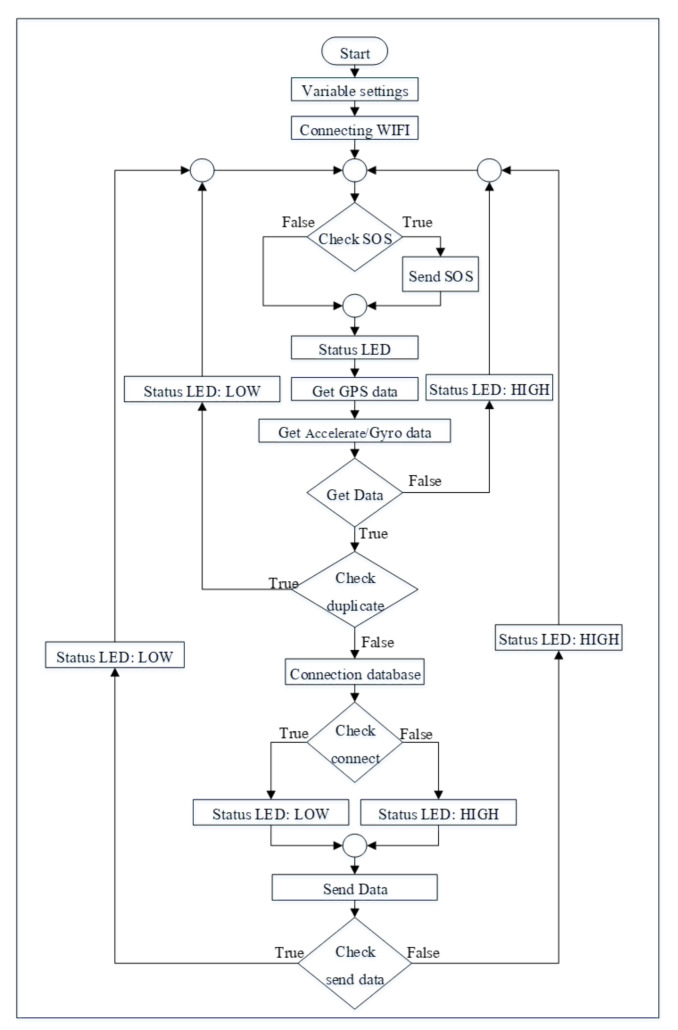
The workflow diagram of the tracking device.

**Figure 6 ijerph-18-12652-f006:**
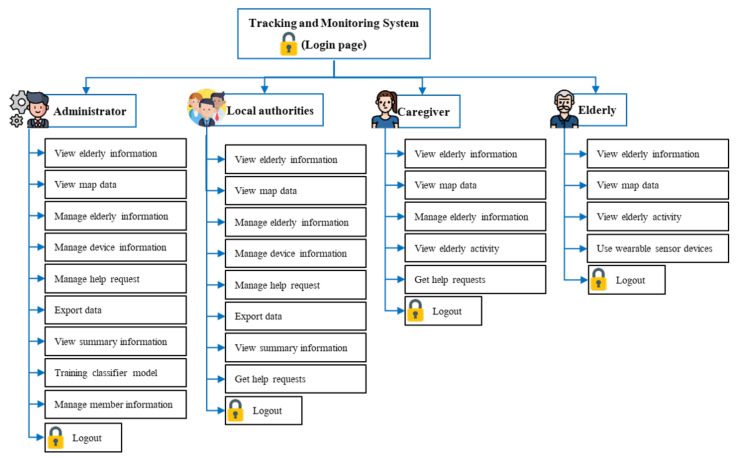
The interface structure design of the system.

**Figure 7 ijerph-18-12652-f007:**
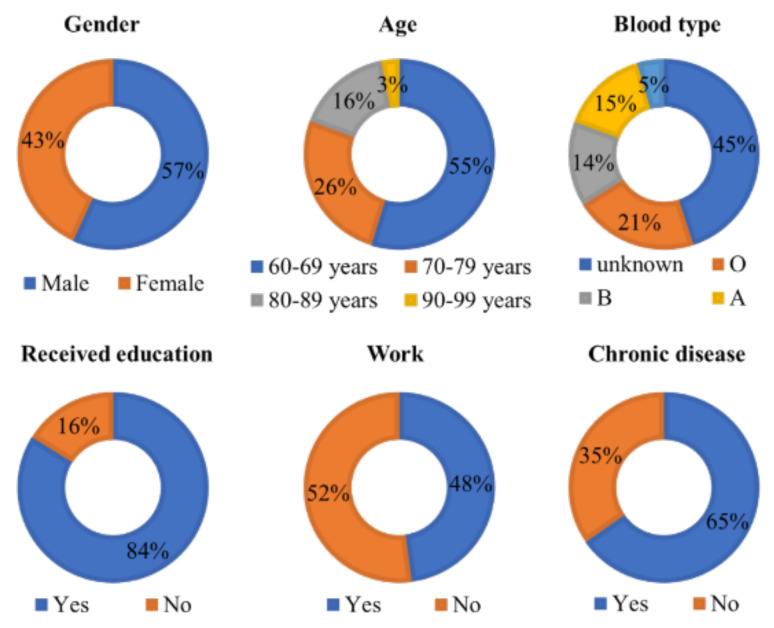
Summary of general information on the population sample.

**Figure 8 ijerph-18-12652-f008:**
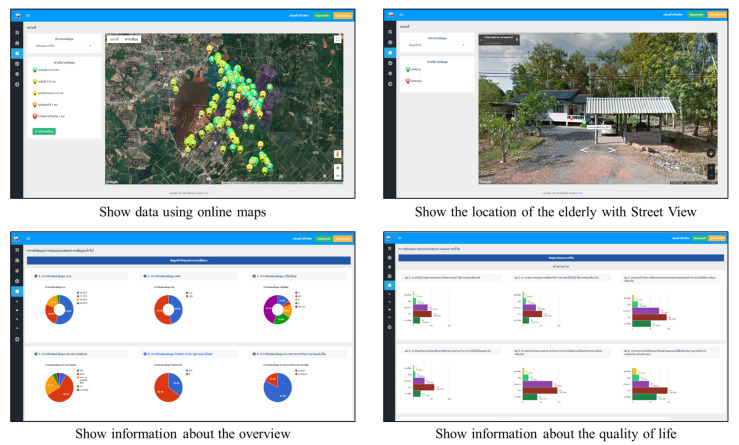
Example screen data displays.

**Figure 9 ijerph-18-12652-f009:**
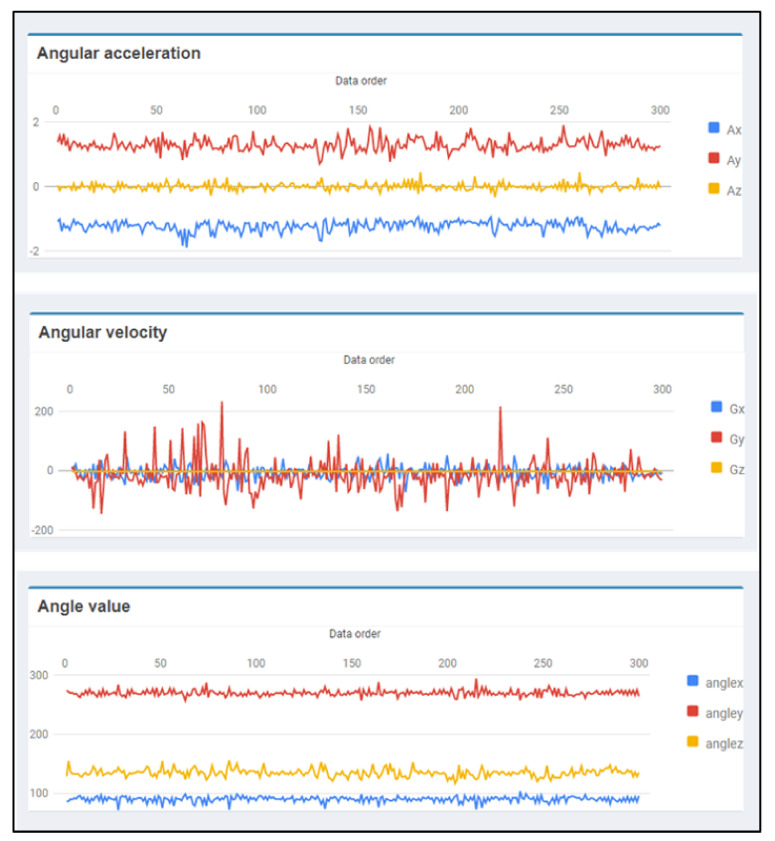
Example data for walking activities from sensors.

**Figure 10 ijerph-18-12652-f010:**
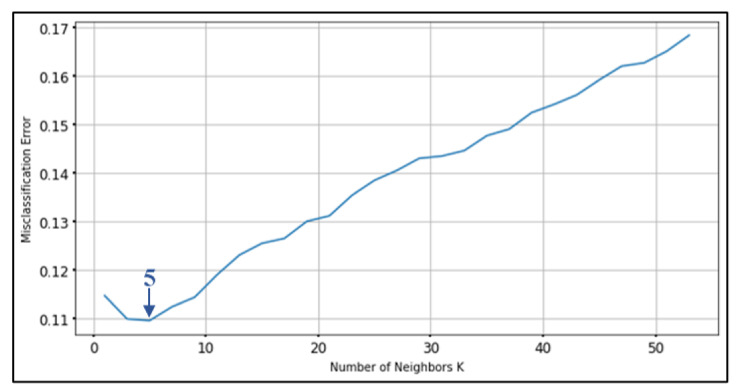
The results of k-fold cross-validation.

**Figure 11 ijerph-18-12652-f011:**
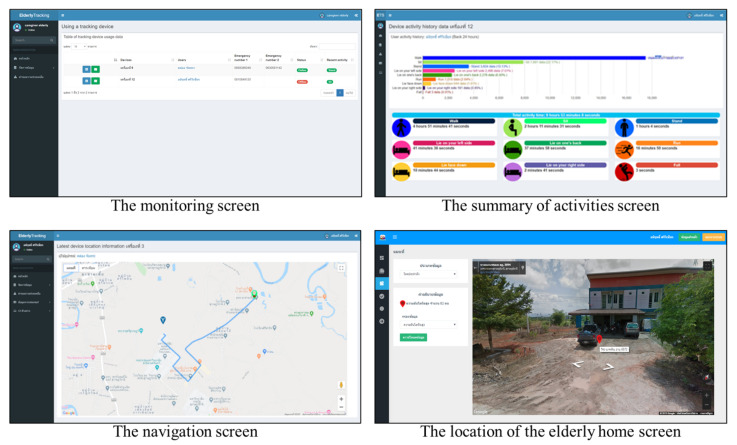
Examples of the activity tracking screen.

**Figure 12 ijerph-18-12652-f012:**
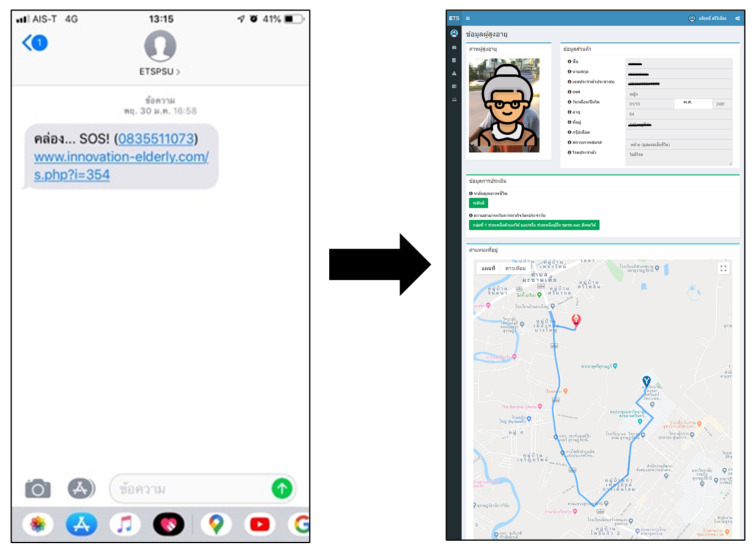
Example of the help message.

**Table 1 ijerph-18-12652-t001:** The collected dataset.

Activity Class	Number of Records
Sitting	1500
Standing	1000
Walking	1500
Running	1000
Falling	1000
Lying on one’s back	1500
Lying face down	1500
Lying on the left side	1000
Lying on the right side	1000

**Table 2 ijerph-18-12652-t002:** Comparing performances by Euclidean and Manhattan distances for k-NN.

The Distance Metric	Performance Result			
	Accuracy	Precision	Recall	F1 Score
Manhattan	89.83%	0.915	0.910	0.912
Euclidean	96.40%	0.918	0.914	0.915

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
