# Peer review of "A Novel Elderly Tracking System Using Machine Learning to Classify Signals from Mobile and Wearable Sensors"

_ijerph, 2021, doi:10.3390/ijerph182312652_

Round 1

Reviewer 1 Report

Technically your IDEA is Good, Justification missing, update references from 2020, 2021 & compare with your Work 

Suggestion 

Ba, T., Li, S. and Wei, Y., 2021. A data-driven machine learning integrated wearable medical sensor framework for elderly care service. Measurement167, p.108383.

Sujaya, B.L. and Bhaskar, R.S., 2021. A Modelling of Context-Aware Elderly Healthcare Eco-System-(CA-EHS) Using Signal Analysis and Machine Learning Approach. Wireless Personal Communications, pp.1-16.

Kumar, V., Badal, N. and Mishra, R., 2021. Elderly fall due to drowsiness: detection and prevention using machine learning and IOT. Modern Physics Letters B35(07), p.2150120.

Piculell, E., Skär, L., Sanmartin Berglund, J., Anderberg, P. and Bohman, D., 2021. A concept analysis of health communication in a home environment: Perspectives of older persons and their informal caregivers. Scandinavian Journal of Caring Sciences35(3), pp.1006-1024.

Al-Khafajiy, M., Baker, T., Chalmers, C., Asim, M., Kolivand, H., Fahim, M. and Waraich, A., 2019. Remote health monitoring of elderly through wearable sensors. Multimedia Tools and Applications78(17), pp.24681-24706.

Author Response

The authors gratefully acknowledge the helpful comments and suggestions of the reviewers, which have improved the presentation.  We appreciate the reviewer’s comments.  The attached files are our point-by-point responses.

Reviewer 2 Report

In this manuscript, the authors develop an elderly tracking system using the integration of multiple technologies combined with machine learning, which covers aspects of activity tracking, geolocation, and personal information in an indoor and outdoor environment. The results  from testing devices and systems in a case study show that the k-nearest neighbor (k-NN) model with k = 5 was the most effective in classifying the 9 activities of the elderly. Overall. The results are interesting. The experiments are sufficient. It is also technically sound. However, this work is not suitable for publication in the current form. Major revisions are required before publications.

1. Why the authors use L2 norm in the KNN algorithm. Is the L1 norm is OK? The authors should add new experiments to compare L1 norm and L2 norm.

2. The KNN algorithm needs to save all training data when it is deployed in a mobile device. Moreover, during test, the L2 norm need to be computed along all training samples. Thus, it is accompanied by a heavy memory and computational overhead. I think that the authors should analyze the inference time or memory cost.

3. Among these machine learning algorithms such as SVM, random forests, why the authors choose the KNN? The authors should rephrase their main contributions in the introduction part.

4. I also recommend the authors to refer to several recent HAR literatures that use machine learning. 

a. Tang, Y., Teng, Q., Zhang, L., Min, F., & He, J. (2020). Layer-wise training convolutional neural networks with smaller filters for human activity recognition using wearable sensors. IEEE Sensors Journal, 21(1), 581-592.

b. Gao, Wenbin, et al. "Deep Neural Networks for Sensor-Based Human Activity Recognition Using Selective Kernel Convolution." IEEE Transactions on Instrumentation and Measurement 70 (2021): 1-13.

Author Response

(The authors gave the same response as above.)

Reviewer 3 Report

Review of IJEnvResearchPubHlth-1449232-Oct2021 “Novel Elderly Tracking System Using Machine Learning to Classify Signals from Mobile and Wearable Sensors”
General Comments
This article describes an integrated sensor-to-activity assessment to improve real-time monitoring of elderly individuals. The determination of the “activity” (including harmful falls) is based on an integrated machine learning model, and the system relies on web access as its technological backbone. It appears that the system is designed adequately, and the overarching description of the overall design and system components is sufficient to understand what has been done.
Unfortunately, however, I find the description of system evaluation confusing, there is a lack of detail about the data used/analysed, the assessment of the accuracy of the system is cursory and uninformative, there is little description of operational deployment of the system, and little consideration about broader applicability. Perhaps most seriously for a scientific article, there is virtually no background about, acknowledgement of, or comparison to existing commercially available systems. Without this, it is impossible to determine what – if anything – is “novel” (as stated in the title) about the system developed. The article seems to be almost a product description for selling systems rather than a scientific article that documents scientific advancement.
Specific Comments
1. The Introduction and Background 2.1 (Human Activity Recognition System) establish a need for wearable sensors, describe the benefits of having such sensors within a health system for the elderly, and describe different system designs. However, there is no statement about what the state of development of wearable sensors for health is. This is necessary to be able to describe why the approach adopted in this research is an advance on what currently exists. And I will add that a quick web search found numerous scientific journal articles on such sensors and systems – and ads selling commercially available systems. The Introduction to this article simply does not identify an existing need or limitation that would be addressed by the authors’ work. It feels very much like the authors’ decided that their machine-learning-based system is the answer to something before deciding what question they are actually asking, and whether or not that question has already been answered.
2. Context is needed for the description of the k-nearest neighbors classification. Presumably kNN is going to be used to analyse sensor data and distinguish between, for example, falls and simply lying down to sleep? But will the model trained be continually recalibrated as more data are collected?
3. The length of the discussion in Section 2.3 explaining Euclidean distance is unnecessary as most people will be familiar with Euclidean distance in two dimensions. It would be more useful to discuss one or more alternatives (such as the Manhattan distance) and describe how it differs from the Euclidean distance both in its calculation and analytically.
4. Figure 1 is a diagram of a system that appears to be realistic. However, is the entire design new? Are only certain components of the design new? Do all components actually exist – e.g., wearable sensors that function indoors and outdoors, and in remote locations?
5. Figure 2 (and elsewhere?): Specifically what data are collected from the sensors? And the information in Figure 2 is based on how many patients? And how much training data were used?
What validation of the machine learning models was done? And while accuracy is important, processing time in human health is also important; is this reported somewhere?
6. A more specific follow up to the previous point is that without a more complete discussion of data, a global accuracy of 90.5% is meaningless. Categorical machine learning models almost always optimize the classification of the most prevalent class – but this occurs at the expense of the minority class(es). At the extreme if only two classes are present and 99% of activities are “walking” and only 1% are “falls,” a global accuracy of 99% can be achieved by simply classifying all observations as “walking.” But the accuracy for “falls” would be 0%. This is not a problem if there is a 50/50 split between walking/falls, but I doubt this is the case in the data used in this study.
7. Lines 218-220. In reference to the need to be connected to the Web, will the system automatically connect to an unknown network? Elderly people sometimes leave their home so would need an automatic connection to the Wifi on a bus, or in a business, or elsewhere.
8. Lines 313-316. So if a sensor cannot receive location data – i.e., GPS data – “home” is assumed to be the location? This is very problematic for indoor use – especially if the elderly person being monitored is mobile and is able to leave home to go shopping, or ride a bus, etc.
9. Figure 8 provides information about the people in the sample. However, it does not provide information about the distribution of activities. In particular, I would assume that “falls” are the rarest activity. Yet they are one of the most critical activities to detect correctly. Were there enough falls in the data set to develop a robust model? What was the accuracy of all activities – not just the global accuracy?
10. Given that the system described is designed for operational use, it would be useful to describe or discuss operational considerations other than sensor weight. What about system cost? Elapsed time from data collection to determination of activity? Bandwidth both for the central server and for individual homes where multiple devices my be used simultaneously for video/audio streaming and other activities?

Author Response

(The authors gave the same response as above.)

Reviewer 4 Report

In this paper, the author proposes a system to track the elderly using the KNN method. The authors focus on the system as a product more than being a research and experiment result. 

The contributions are not clearly addressed, they can be improved.

The results are not explained properly. 

The paper is missing details regarding getting 90.5% accuracy. What methods are used to find the accuracy?

Author Response

(The authors gave the same response as above.)

Reviewer 5 Report

algorithm should be table not figure

privacy  

Author Response

The authors gratefully acknowledge the helpful comments and suggestions of the reviewers, which have improved the presentation.  We appreciate the reviewer’s comments.  However, we will edit in the manuscript.

Round 2

Reviewer 1 Report

The authors have satisfactorily addressed the questions. 

Reviewer 2 Report

The authors have satisfactorily answered my concerns, and I can recommend it for a publication.

Reviewer 4 Report

Thank you for responding to all comments and making all necessary changes.